# Expression of GADD45G and CAPRIN1 in Human Nucleus Pulposus: Implications for Intervertebral Disc Degeneration

**DOI:** 10.3390/ijms24065768

**Published:** 2023-03-17

**Authors:** Koki Kawaguchi, Koji Akeda, Junichi Yamada, Takahiro Hasegawa, Norihiko Takegami, Tatsuhiko Fujiwara, Akihiro Sudo

**Affiliations:** Department of Orthopaedic Surgery, Graduate School of Medicine, Mie University, Tsu 514-8507, Japan

**Keywords:** intervertebral disc, degeneration, cell cycle, nucleus pulposus, human, GADD45G, CAPRIN1

## Abstract

Marked cellular changes occur in human intervertebral disc (IVD) degeneration during disc degeneration with biochemical changes. Genome-wide analysis of the DNA methylation profile has identified 220 differentially methylated loci associated with human IVD degeneration. Among these, two cell-cycle–associated genes, growth arrest and DNA damage 45 gamma (GADD45G) and cytoplasmic activation/proliferation-associated protein-1 (CAPRIN1), were focused on. The expression of GADD45G and CAPRIN1 in human IVDs remains unknown. We aimed to examine the expression of GADD45G and CAPRIN1 in human nucleus pulposus (NP) cells and evaluate those in human NP tissues in the early and advanced stages of degeneration according to Pfirrmann magnetic resonance imaging (MRI) and histological classifications. Human NP cells were cultured as monolayers after isolation from NP tissues by sequential enzyme digestion. Total RNA was isolated, and the mRNA expression of GADD45G and CAPRIN1 was quantified using real-time polymerase chain reaction. To examine the effects of pro-inflammatory cytokines on mRNA expression, human NP cells were cultured in the presence of IL-1β. Protein expression was evaluated using Western blotting and immunohistochemistry. GADD45G and CAPRIN1 expression was identified in human NP cells at both mRNA and protein levels. The percentage of cells immunopositive for GADD45G and CAPRIN1 significantly increased according to the Pfirrmann grade. A significant correlation between the histological degeneration score and the percentage of GADD45G-immunopositive cells was identified, but not with that of CAPRIN1-immunopositive cells. The expression of cell-cycle-associated proteins (GADD45G and CAPRIN1) was enhanced in human NP cells at an advanced stage of degeneration, suggesting that it may be regulated during the progression of IVD degeneration to maintain the integrity of human NP tissues by controlling cell proliferation and apoptosis under epigenetic alteration.

## 1. Introduction

Low back pain (LBP) is one of the most common health concerns in patients, ranging from children to the elderly, and it affects daily activities and quality of life [1]. Recent epidemiological and clinical studies have reported that intervertebral disc (IVD) degeneration is associated with the occurrence of LBP [2,3,4,5]. Clinically, IVD degeneration is well assessed using magnetic resonance imaging (MRI), and its severity and characteristic findings are reported to be associated with discogenic LBP [6,7].

IVDs are complex structures that consist of a thick outer ring of fibrous cartilage termed the annulus fibrosus (AF), which surrounds a more gelatinous core known as the nucleus pulposus (NP); the NP is sandwiched inferiorly and superiorly by cartilage end-plates [8]. IVD degeneration is defined as “the structural and functional failure of the disc as a result of aberrant, pathological cellular and extracellular matrix (ECM) changes” [9]. Although the precise causes remain to be elucidated, IVD degeneration is considered to be related to genetic factors, environmental factors, sex, age, and poor nutrition [10].

The progression of IVD degeneration is suggested to be a consequence of an imbalance between the anabolism and catabolism of the ECM [11]. The biochemical characteristics of IVD degeneration, especially NP degeneration, include changes in ECM molecules (mainly, the loss of proteoglycan and water content in the NP). Along with biochemical changes, marked cellular changes occur in the human NP during the progression of IVD degeneration.

Cells in the human nucleus are initially notochordal but are gradually replaced during childhood by rounded cells resembling the chondrocytes of articular cartilage, which is considered to be the first sign of disc degeneration [12]. With increasing degeneration, the proportion of single cells in the human NP continuously decreases and thereafter becomes extremely low in adulthood. Simultaneously, cell clusters increase, leading to cell-mediated tissue remodeling with the progression of disc degeneration.

A previous study evaluated the association between DNA methylation and disc degeneration by genome-wide association analysis of human NP tissues and identified 220 differentially methylated loci associated with human IVD degeneration [13]. Among the differentially methylated genes in the advanced stages of disc degeneration, those related to the cell cycle were extracted. Thereafter, two pivotal genes functionally associated with the cell cycle, one for cell death (apoptosis) and the other for cell proliferation, were identified for investigation in this study. Growth arrest and DNA damage 45 gamma (GADD45G) has been implicated in regulating cell survival, apoptosis, senescence, cell cycle arrest, and genomic stability (see review in [14,15,16]). Cytoplasmic activation/proliferation-associated protein-1 (CAPRIN1) is an RNA-binding protein that has been identified as a promoter of cell proliferation [17] and participates in the regulation of post-transcriptional responses to stress [18,19,20]. Therefore, we hypothesized that these two cell-cycle–associated genes are expressed in the human NP and play a role in cellular changes in the process of disc degeneration.

This study aims to (1) examine the expression of GADD45G and CAPRIN1 in human NP cells and (2) compare the expression of GADD45G and CAPRIN1 in human NP tissues at different stages of degeneration, as evaluated by Pfirrmann MRI classification and the newly modified histological human NP classification score.

## 2. Results

### 2.1. Histological Classification of Human NP Tissues at Different Stages of Degeneration

In the hematoxylin and eosin (H-E) staining samples, a number of chondrocyte-like single cells was observed in the NP of Pfirrmann grade 2 (Figure 1a,d); however, as degeneration progressed, the number of single cells decreased and cell cluster formations were observed in the Pfirrmann grade 3 (Figure 1b,e) and 4 samples (Figure 1c,f). In the Safranin-O staining samples, a red-stained matrix was evenly distributed in the NP of Pfirrmann grade 2 (Figure 2a,d). On the other hand, the intense staining of Safranin-O around the cell clusters (pericellular matrix) was identified in Pfirrmann grade 3 (Figure 2b,e) and 4 samples (Figure 2c,f). The staining property of Safranin-O was remarkably decreased, and micro-fissures were observed in the Pfirrmann grade 4 sample (Figure 2c,f).

In all NP samples (n = 30), the mean total histological score was 2.50 ± 1.3. In the subclass analysis, the mean histological score of “cellularity” was 0.76 ± 0.62, “matrix” was 1.00 ± 0.58, and “matrix staining” was 0.73 ± 0.58. The total histological score in the Pfirrmann grade 4 group was significantly higher than that in the grade 2 group (*p* < 0.01; grade 2 (n = 4), 1.50 ± 0.57; grade 3 (n = 11), 1.90 ± 1.44; grade 4 (n = 15), 3.20 ± 1.08) (Figure 3a). The histological score of “cellularity” in the Pfirrmann grade 4 group was significantly higher than in the grade 2 group (*p* < 0.01; grade 2: 0, grade 3: 0.72 ± 0.64, grade 4: 1.00 ± 0.53) (Figure 3b). The score of “matrix” in the Pfirrmann grade 4 group was significantly higher than that in the grade 3 group (*p* < 0.05; grade 2: 0.75 ± 0.50, grade 3: 0.63 ± 0.50, grade 4: 1.33 ± 0.48) (Figure 3c). There was no significant difference in the histological score of “matrix staining” among the groups (*p* = 0.39; grade 2: 0.75 ± 0.50, grade 3: 0.54 ± 0.68, grade 4: 0.86 ± 0.51) (Figure 3d).

### 2.2. Correlation between Pfirrmann MRI Classification and Histological Classification Scores

The Pfirrmann MRI classification score was positively correlated with the total histological classification score (*r* = 0.54, *p* < 0.01). The subclass analysis showed that the Pfirrmann MRI classification score was significantly correlated with the histological classification score of “cellularity” (*r* = 0.49, *p* < 0.01) and “matrix” (*r* = 0.53, *p* < 0.01); however, no significant correlation with “matrix staining” was identified.

### 2.3. Total Cell Number by MRI Grade Classification

The total number of cells in each view captured by microscopy was counted and compared by Pfirrmann MRI classification. No significant difference in the total number of cells was found among the three grades (grade 2: 14.6 ± 10.5, grade 3: 15.8 ± 16.7, grade 4: 15.5 ± 1.4, Figure 4).

### 2.4. Gene Expression of GADD45G and CAPRIN1 and Effect of Interleukin-1β Stimulation

The mRNA expressions of GADD45G and CAPRIN1 were clearly identified in human NP cells (Figure 5). IL-1β increased the mRNA expression of GADD45G in a dose-dependent manner (IL-1β 0.1 [ng/mL]: 1.62 ± 1.30, IL-1β 1.0: 4.59 ± 5.56, IL-1β 10: 10.08 ± 10.44-fold vs. IL-1β 0); however, it did not reach statistical significance (IL-1β 10 ng/mL group vs. IL-1β 0 control group, *p* = 0.053, Figure 5a). Similarly, an increase of relative mRNA expression of CAPRIN1 stimulated by IL-1β was found (IL-1β 0.1 [ng/mL]: 0.96 ± 0.93, IL-1β 1.0: 1.38 ± 1.17, IL-1β 10: 1.59 ± 1.07-fold vs. IL-1β 0). However, no significant differences were identified among the groups (IL-1β 0 control vs. IL-1β 10 ng, *p* = 0.50, Figure 5b).

### 2.5. Western Blot Analysis of Human NP Cells for GADD45G and CAPRIN1

Western blot analysis identified a single band directed against GADD45G (60 kDa) and CAPRIN1 (116 kDa) in the protein extracts from NP cells. β-actin expression was clearly identified in the NP cells. The relative intensity of GADD45G normalized to β-actin was 0.81, and that of CAPRIN1 was 0.63 (Figure 6).

### 2.6. Immunohistochemical Expression of GADD45G in Human NP Tissues at Different Stages of Degeneration

Immunoreactivity directed against GADD45G was clearly found in human NP cells in NP tissues for all Pfirrmann MRI classifications (Figure 7). Immunoreactivity against GADD45G was observed around the nuclei of chondrocyte-like cells in the NP region of the MRI grade 2 sample (Figure 7a). Immunopositive cells with intense staining were identified in the cells of cluster-forming MRI grade 3 and 4 samples (Figure 7b,c). No immunoreactivity was found in the isotype controls (Figure 7d).

The percentage of GADD45G-immunopositive cells in the NP lesions in the Pfirrmann grade 4 group (74.2% ± 11.3%) was significantly higher than that in the grade 2 (45.7% ± 15.8%, *p* < 0.01) and grade 3 (61.5% ± 17.1%, *p* < 0.05) groups (Figure 7e). The percentage of 2+ positive cells in the Pfirrmann grade 4 group (49.2% ± 15.2%) was significantly higher than that in the Pfirrmann grade 2 (23.5% ± 16.0%, *p* < 0.01) and 3 (32.1% ± 17.7%, *p* < 0.05) groups, respectively (Figure 7f).

The Pfirrmann MRI grade was positively correlated with the percentage of GADD45G-immunopositive cells (*r* = 0.55, *p* < 0.01).

### 2.7. Immunohistochemical Expression of CAPRIN1 in Human NP Tissues at Different Stages of Degeneration

CAPRIN1 immunopositive cells were also found in the cytoplasm of chondrocyte-like cells of the NP regions at all Pfirrmann MRI classifications (Figure 8a–c). No immunoreactivity was found in the isotype controls (Figure 8d). The percentage of CAPRIN1-immunopositive cells in the Pfirrmann grade 3 (84.0% ± 4.0%) and grade 4 (85.9% ± 9.2%) groups was significantly higher than that in the grade 2 (71.3% ± 7.4%) group (*p* < 0.05, *p* < 0.01, respectively) (Figure 8e). The percentage of 1+ positive cells in the Pfirrmann grade 4 group (72.1% ± 12.1%) was significantly higher than those in the Pfirrmann grade 2 (55.3% ± 6.6%) and 3 (54.4% ± 13.2%) groups, respectively (*p* < 0.05, *p* < 0.01). The percentage of 2+ positive cells in the Pfirrmann grade 3 group (29.5% ± 13.3%) was significantly higher than those in the Pfirrmann grade 4 group (13.7% ± 12.9%, *p* < 0.01) (Figure 8f). The Pfirrmann MRI grade was positively correlated with the percentage of CAPRIN1-immunopositive cells (*r* = 0.50, *p* < 0.01).

### 2.8. Correlation between Percentage of Immunopositive Cells and Histological Classification Score

The percentage of GADD45G-immunopositive cells positively correlated with the total histological score (*r* = 0.47, *p* < 0.01). The subclass analysis showed that the percentage of GADD45G-immunopositive cells was significantly correlated with the histological scores of “matrix” (*r* = 0.39, *p* < 0.05) and “matrix staining” (*r* = 0.40, *p* < 0.05); however, no significant correlation with that of “cellularity” was found (Figure 9). In contrast, there were no significant correlations between the percentage of CAPRIN1-immunopositive cells and the histological score (Figure 10a) or its subtype score (Figure 10b–d).

## 3. Discussion

This study shows, for the first time, that GADD45G and CAPRIN1 are expressed in human NP cells at both mRNA and protein levels. GADD45G and CAPRIN1 expression in NP cells tend to be stimulated by pro-inflammatory cytokines (IL-1β), and their expression is significantly upregulated in degenerated IVD tissues. The expression of both GADD45G and CAPRIN1 is intermediately correlated with Pfirrmann MRI classification, and the expression of GADD45G is also significantly correlated with the histological classification score of the NP tissue.

MRI is considered the best imaging instrument for evaluating IVD degeneration. The most widely known classification of IVD degeneration was reported by Pfirrmann et al. [21]; it evaluates the disc signal intensity, disc structure, distinction between the nucleus and annulus, and disc height for classifying the degree of disc degeneration into five grades with adequate inter- and intra-observer agreement [21,22]. Thus, the Pfirrmann classification is graded to reflect whole-disc degeneration, including NP and AF. Previous studies have shown that the Pfirrmann MRI classification in animal IVDs has a significant correlation with the histological score of disc degeneration [23,24,25], in particular using the degeneration score by Boos classification [26]; however, few studies have evaluated these correlations in human IVDs [27].

Human IVD tissues obtained during spinal surgeries, particularly lumbar interbody fusion surgeries, largely contain NP tissues. Therefore, macroscopically identified human NP tissues were used in this study. The histology of the human NP tissues was evaluated using the new histological scoring system by Rutges et al. [28] with modifications, in particular, by adopting three items associated with the histology of NP. It has been reported that the Rutges scoring system has been biochemically validated with high intra- and inter-observer reliability [28]; however, the association with MRI-graded human disc degeneration remains unknown. The results of this study revealed that the Pfirrmann classification was significantly correlated with the total histological grading scores and subclass grading scores, except for “NP matrix staining”. Our results suggest that our modified Rutges classification for human NP shows a statistically significant correlation with MRI-graded disc degeneration evaluated using the Pfirrmann classification [21].

GADD45G, a member of the GADD45 family, encodes a small (18 kDa) protein that negatively regulates cell growth. Importantly, GADD45 proteins can form homo- and/or hetero-oligomers with other family members [29] and play a role by interacting with cell-cycle-related proteins. In our study, Western blot analysis showed a single band of GADD45G at 60 kDa in human NP samples, which reflects the oligomerization of GADD45G with other GADD proteins and/or cell-cycle-related proteins. Previous reports have shown that GADD45G interacts with and inhibits the kinase activity of the Cdk1/CyclinB1 complex [30], which plays a key role in the progression from the G2 to M phase of the cell cycle [31].

The expression of GADD45 family members, including GADD45G, is known to be induced by various physiological stresses, including irradiation, ultraviolet radiation, and inflammatory cytokines [14,32,33]. Our results showed that the mRNA expression of GADD45G was upregulated by stimulation with the pro-inflammatory cytokine IL-1β. Furthermore, the percentage of GADD45G-immunopositive cells in the human NP of the Pfirrmann grade 4 samples was significantly higher than that of the grade 2 and 3 samples. These results suggest that GADD45G expression is upregulated in human NP at an advanced stage of degeneration, where the aberrant expression of pro-inflammatory cytokines, such as IL-1β or TNF-α, is found.

CAPRIN1 was purified from activated T-lymphocytes, as reported by Grill et al. [17]. CAPRIN1 is a ubiquitously expressed RNA-binding protein that participates in the regulation of cell-cycle-control–associated genes in the G1 to S phase transition [19,34,35]. The expression of CAPRIN1 was reported to be high in the thymus and spleen and low in tissues with a low proportion of dividing cells, such as the kidney or muscle [17]. Importantly, CAPRIN1 was reported to be a core nucleating component of stress granules, which are dense aggregates in the cytosol composed of proteins and RNAs that appear when the cell is under stress [36].

Gene expression analysis by real-time PCR in this study did not show a significant association between CAPRIN1 expression and the pro-inflammatory cytokine stimulus IL-1β, which plays a major role in matrix degradation and pain generation by promoting the expression of degradative enzymes, such as MMP-3 or -13 and ADAMTS-4 or -5 [37], and pain-related molecules, such as nerve growth factor [38] and glial-cell-line-derived neurotrophic factor [39]. The regulatory mechanism of CAPRIN1 expression remains largely unknown; however, it is speculated that CAPRIN1 and stress granules can regulate the cellular post-transcriptional response to various types of stress, such as osmotic pressure or nitric oxide [40,41].

Histological evaluation in this study showed that the percentage of CAPRIN1 immunopositive cells in Pfirrmann grade 3 or 4 samples was higher than that in grade 2 samples. Furthermore, a higher percentage of strongly immunopositive cells was observed in the grade 3 samples than in the grade 4 samples. In the process of disc degeneration, IVD cells attempt to restore the damaged matrix by forming cell clusters, and a subsequent reduction in viable cell numbers appears as degeneration advances [42]. The change in the number and intensity of CAPRIN1-immunopositive cells among the different grades of Pfirrmann MRI classifications may reflect a change in cell proliferative activity during the progression of disc degeneration. The results of this study may explain the mechanism of disc degeneration from the viewpoint of cell proliferation; however, further investigation is needed to elucidate the CAPRIN1 response to various stressors that have been reported to be associated with disc degeneration, such as oxidative stress [43].

GADD45G and CPRIN1 are representative hypermethylated genes in the advanced degenerated IVD group and are associated with cell cycling. Theoretically, when methylation is located in gene promoter and enhancer regions, DNA methylation typically acts to silence genes, whereas methylation in gene body regions usually induces enhanced gene expression [44]. However, recent genome-wide studies of DNA methylation have reported that gene expression differs significantly depending on the methylation site of the promoter lesion (5′UTR or 3′UTR side) [45]. Previous genome-wide DNA methylation analysis has shown that GADD45G and CAPRIN1 are representative genes in which core promoter lesions were hypermethylated in the NP in the advanced stage of disc degeneration [13]; however, the percentage of GADD45G- and CAPRIN1-immunopositive cells was higher in the human NP with advanced stages of disc degeneration. Therefore, authors have speculated that the epigenetic regulation of gene transcription is not only confined to DNA methylation but is also intricately cooperative, including with the influence of chromatin variation and noncoding RNA [46].

There were several limitations in this study. First, mRNA was not extracted from the human NP tissues; therefore, the mRNA expression of GADD45G and CPRIN1 was not quantitatively evaluated. Second, our study representatively focused on GADD45G and CAPRIN1, which were differentially methylated in advanced-stage degenerated IVD tissues [13]; however, other cell-cycle-related molecules may also be responsive to the progression of IVD degeneration. Thirdly, the characterization of the cellular phenotype in the human NP is important for the immunohistological evaluation of the cell-cycle-associated proteins; however, it was not evaluated in this study. NP chondrocyte markers such as Krt19, Pax1, and FoxF1 [47] should be evaluated in future studies.

## 4. Materials and Methods

This study was approved by the Institutional Clinical Research Ethics Review Committee of Mie University Hospital (approval number: H2022-178) and performed in accordance with the Declaration of Helsinki. Written informed consent was obtained from all patients.

### 4.1. Human NP Tissues and Cell Isolation

Human IVDs were obtained from surgical specimens (five women, 50–75 years of age (mean 67 years) and with Pfirrmann MRI grades 4 [21]). NP tissues were macroscopically separated from other structures of human IVD tissues, including the AF and cartilaginous end-plates. Human NP cells were cultured in a monolayer after isolation from NP tissues through sequential enzyme digestion, as previously reported [48]. Briefly, following 0.4% pronase and 0.025% collagenase P digestion, the cells were washed with Dulbecco’s modified Eagle’s medium and Ham’s F-12 medium (DMEM/F12; Gibco, Palo Alto, CA, USA), and cultured in a monolayer at 4.0 × 10^4^ cells/mL with 5% CO_2_ and 95% air in complete medium (DMEM/F12 containing 10% fetal bovine serum, 25 μg/mL ascorbic acid, 10,000 units/mL penicillin, and 10,000 μg/mL streptomycin) (Figure 11). The medium was changed every three days. Primary cultured cells were used in all experiments conducted in this study. The cells in six-well plates were pre-cultured to 80% confluence (approximately 14 d) in all sets of experiments.

### 4.2. Western Blotting

Cell lysates (containing 20 μg of protein) of monolayer-cultured NP cells were analyzed using Western blotting under reducing conditions, as previously reported [49]. For GADD45G-immunoblotting, the cell lysates were treated with chondroitinase-ABC, degrading the proteoglycans’ chondroitin sulfate chains. Immunostaining was performed using a mouse monoclonal antibody raised against GADD45G (OTI2F12, 1:4000; Novus Biologicals, Littleton, CO, USA) and CAPRIN1 (15112-1-AP, 1:500; Proteintech, Manchester, UK), diluted with 5% skim milk in Tris buffered saline (TBS). β-actin served as a loading control for the Western blot assay. The densitometry analysis was performed using Image J software (version 1.53 m; National Institutes of Health, Bethesda, MD, USA).

### 4.3. RNA Isolation

Total RNA was isolated from human NP cells in monolayer culture using the ISOGEN PB kit (NipponGene, Toyama, Japan) according to the manufacturer’s instructions. Total RNA was reverse-transcribed using the first strand cDNA synthesis kit (Roche Applied Science, Mannheim, Germany) with a DNA thermal cycler (Veriti, Applied Biosystems, Foster City, CA, USA) according to the manufacturer’s protocol.

### 4.4. Quantitative Real-Time Polymerase Chain Reaction (PCR)

The expression levels of GADD45G (Hs02566147 s1, TaqMan Gene Expression Assay; Applied Biosystems) and CAPRIN1 (Hs00195416 m1; Applied Biosystems) were quantified via real-time PCR with TaqMan gene expression assays (Applied Biosystems) using the primer pairs for TaqMan genomic assays. The assay was calibrated using 18S RNA (Hs99999901 s1) as an internal control. To determine the expression levels of GADD45G and CAPRIN1, the resultant cDNA (three replicates) was amplified for the target genes. The cycle used a 15 s denaturation at 95 °C and 1 min annealing and extension at 60 ℃, utilizing the ABI PRISM 7000 sequence detection system (Applied Biosystems). The relative expression levels of GADD45G and CAPRIN1 were calculated using the comparative threshold method [50].

### 4.5. Effect of Interleukin-1β on the Gene Expression of GADD45G and CAPRIN1

To examine the effect of pro-inflammatory cytokines on the mRNA expression of GADD45G and CAPRIN1, human NP cells were cultured in the presence of IL-1β (0.1, 1.0, and 10 ng/mL) for 48 h after serum starvation. The mRNA expression levels of GADD45G and CAPRIN1 were quantified as described above.

### 4.6. Histological Grading of Human NP Tissues

Human NP tissues obtained from spine surgeries were divided into three groups using Pfirrmann MRI classification (14 men, 16 women, 34–85 years old, mean age: 64.5 years old, grade 2: n = 4, grade 3: n = 11, grade 4: n = 15) (Table 1), according to the extent of disc degeneration evaluated via MRI. The samples were fixed in 4% paraformaldehyde for 48 h at 4 ℃ and embedded in paraffin. The sections (5 μm) were stained with H-E, Safranin-O, and fast green. The histological grading of human NP tissues was evaluated based on Rutges’s classification [28] with modifications. NP tissues were evaluated in three subcategories. Each item was graded as 0, 1, or 2 on the H-E sections for cellularity NP and matrix NP and Safranin-O sections for NP matrix staining. Cellularity NP: 0: Normal cellularity, 1: Mixed cellularity, 2: Mainly clustered cellularity; Matrix NP: 0: Well-organized structure of nucleus matrix, 1: Partly disorganized structure of nucleus matrix, 2: Complete disorganization and loss of nucleus matrix; NP matrix staining: 0: Intense staining (red stain dominates), 1: Reduced staining (mixture of red and slight green staining), 2: Faint staining (increased green staining). The total score of the modified classification is the sum of the three different scoring items of NP tissues, resulting in a minimum score of 0 points for a completely healthy NP and a maximum of 6 points for an entirely degenerated NP.

### 4.7. Immunohistochemistry of Human NP Tissues

For the immunohistochemistry of GADD45G and CAPRIN1, the human NP tissues used in the histological analysis were analyzed. For the immunostaining of GADD45G, after epitope retrieval with citrate buffer (pH 6.0), sections were incubated with a primary anti-GADD45G mouse monoclonal antibody (Novus Biologicals, Littleton, CO, USA). For the staining of CAPRIN1, after epitope retrieval using proteinase K treatment, the sections were incubated with primary anti-CAPRIN1 rabbit polyclonal antibody (Proteintech, Manchester, UK) diluted with 1% bovine serum albumin (BSA) in phosphate buffered saline (PBS). The primary antibody was visualized using the Histofine Simple Stain MAX-PO (MULTI) kit (Nichirei Bioscience, Tokyo, Japan), according to the manufacturer’s instructions, with some modifications. Peroxidase activity was detected using 3,3′-diaminobenzidine tetrahydrochloride (DAB; Dojindo, Toyama, Japan). The sections were counterstained with Mayer’s hematoxylin. The isotype control was processed using mouse IgG. Five views of each section of the NP area at 200× magnification were randomly captured, and immunopositive or -negative cells were manually counted using conventional microscopy (OLYMPUS BX53, Tokyo, Japan). Immunopositive cells were classified as slightly positive (1+) or strongly positive (2+) according to the staining area and intensity (Table 2).

### 4.8. Statistical Analysis

Data are expressed as the mean ± standard deviation. One-way analysis of variance (ANOVA) was used to assess the effects of the culture conditions in vitro. Post-hoc analyses were performed using Fisher’s least significant difference (LSD). Statistical differences in histological grading scores by Pfirrmann MRI classification were determined using the Kruskal–Wallis test. Statistical correlations between Pfirrmann MRI classification and histological classification scores and between the percentage of immunopositive cells and Pfirrmann MRI classification score or histological classification score were evaluated using Spearman’s correlation test. All statistical analyses were performed using IBM SPSS Statistics software (version 28.0; IBM Japan, Tokyo, Japan). Statistical significance was set at *p* < 0.05.

## 5. Conclusions

Our study showed, for the first time, that GADD45G and CAPRIN1 were expressed in human NP cells at both the mRNA and protein levels. Immunohistochemical analysis showed the enhanced expression of GADD45G and CAPRIN1 in advanced stages of degenerated NP tissues. We speculated that the expression of these two cell-cycle-related molecules may be regulated during IVD degeneration progression to maintain the integrity of human NP tissues by controlling cell proliferation and apoptosis (cell death) under epigenetic alteration.

## Figures and Tables

**Figure 1 ijms-24-05768-f001:**
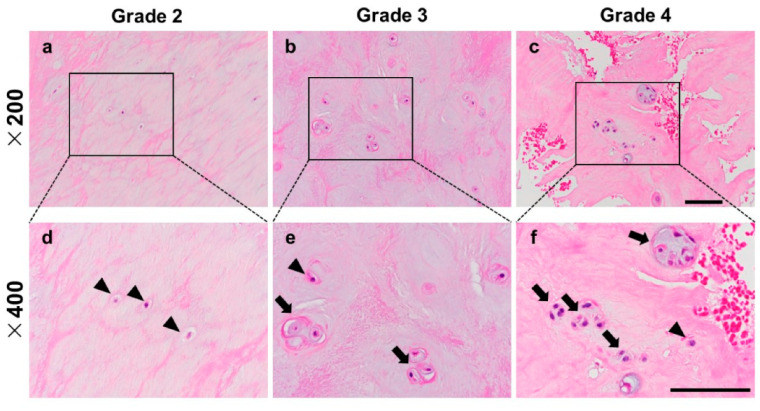
Representative histology of the human nucleus pulposus (NP) stained by H-E. NP tissue classified in Pfirrmann grade 2 (**a**,**d**), Pfirrmann grade 3 (**b**,**e**), Pfirrmann grade 4 (**c**,**f**). The arrowhead indicates single cells, and the arrow indicates cell clusters. (**a**–**c**): ×200 magnification, (**d**–**f**): ×400 magnification. Scale bar: 100 μm.

**Figure 2 ijms-24-05768-f002:**
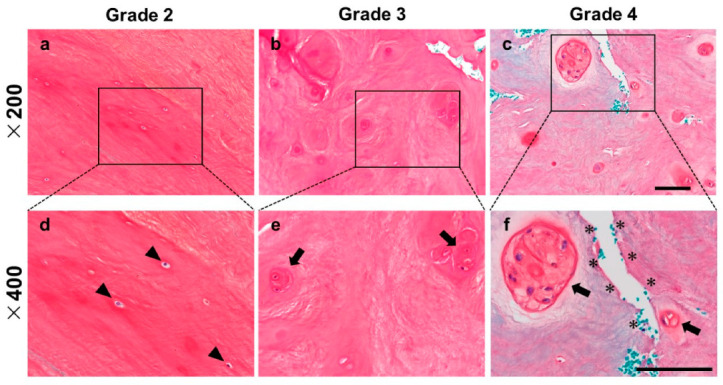
Representative histology of the human NP stained by Safranin-O. NP tissue classified in Pfirrmann grade 2 (**a**,**d**), Pfirrmann grade 3 (**b**,**e**), Pfirrmann grade 4 (**c**,**f**). The arrowhead indicates single cells, the arrow indicates cell clusters, and the asterisk indicates micro-fissures. (**a**–**c**): ×200 magnification, (**d**–**f**): ×400 magnification. Scale bar: 100 μm.

**Figure 3 ijms-24-05768-f003:**
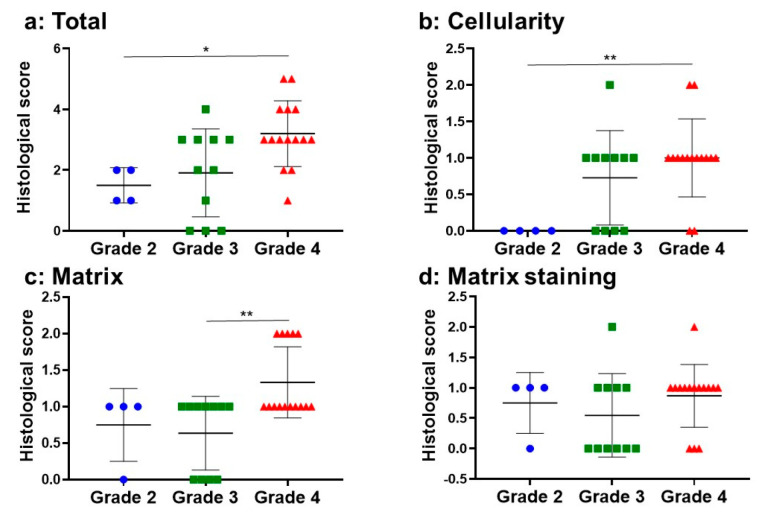
Histological classification of human NP tissues by Pfirrmann classification. (**a**): total, (**b**): cellularity, (**c**): matrix, (**d**): matrix staining. Grade 2 (n = 4), grade 3 (n = 11), grade 4 (n = 15). * *p* < 0.05, ** *p* < 0.01, Kruskal–Wallis test.

**Figure 4 ijms-24-05768-f004:**
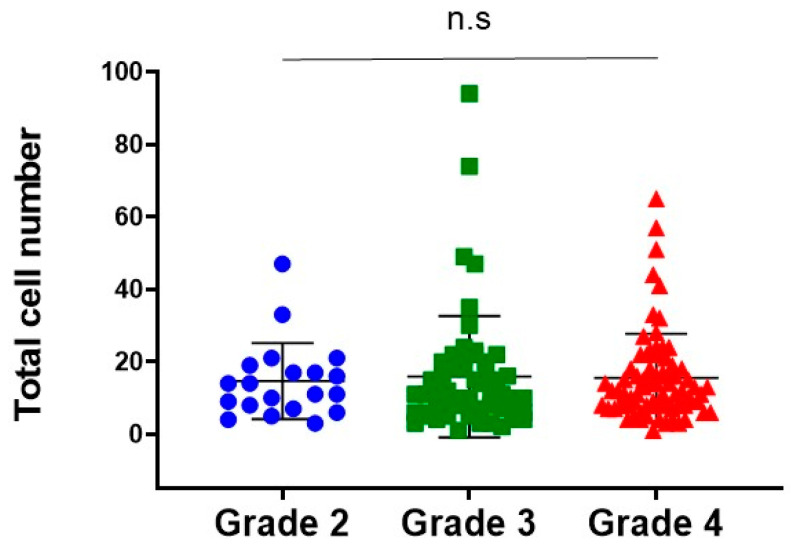
Total cell number by MRI grade classification. n.s not significant.

**Figure 5 ijms-24-05768-f005:**
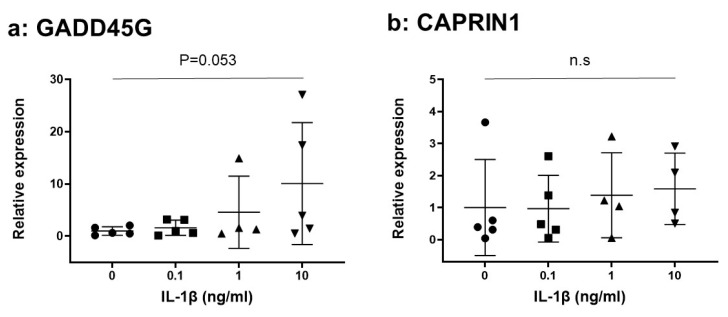
Effect of interleukin-1 beta (IL-1β) on the mRNA expression of GADDD45G (**a**) and CAPRIN1 (**b**) in human nucleus pulposus (NP) cells (n = 5). n.s not significant.

**Figure 6 ijms-24-05768-f006:**
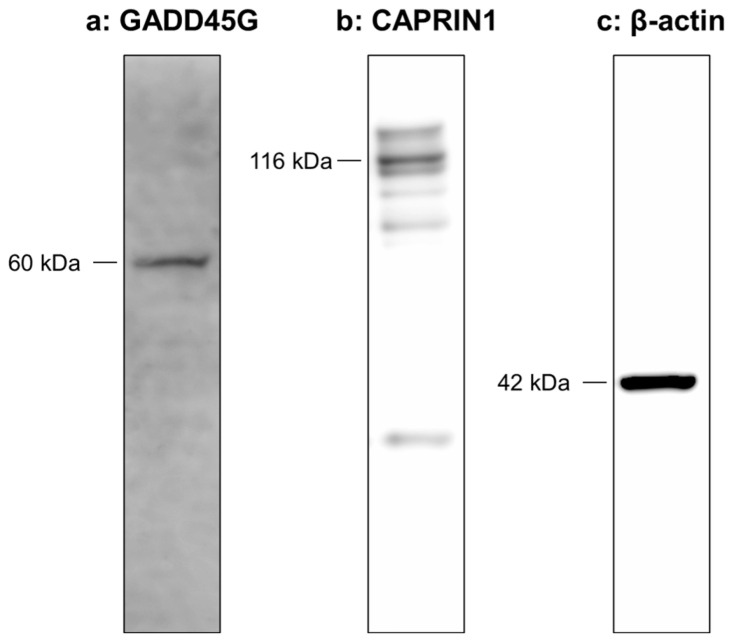
Western blotting of GADD45G (**a**) and CAPRIN1 (**b**) expression in cultured human nucleus pulposus (NP) cells. β-actin (**c**) served as a loading control.

**Figure 7 ijms-24-05768-f007:**
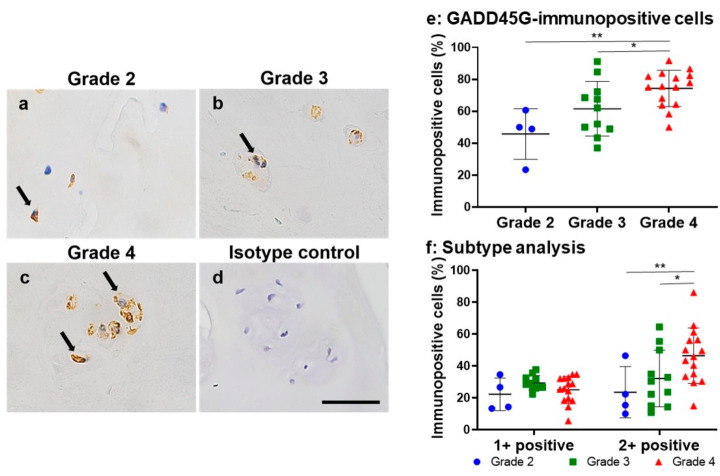
Immunohistochemical staining for GADD45G (**a**–**c**) in human nucleus pulposus (NP) tissues. Pfirrmann grade 2 (**a**), grade 3 (**b**), grade 4 (**c**), and isotype control (**d**). Scale bar: 50 μm (×400 magnification). Percentage of immunopositive cells for GADD45G (**e**) and the subtype analysis by the intensity of immunostaining (**f**) in human nucleus pulposus (NP) tissues by Pfirrmann classification. * *p* < 0.05, ** *p* < 0.01.

**Figure 8 ijms-24-05768-f008:**
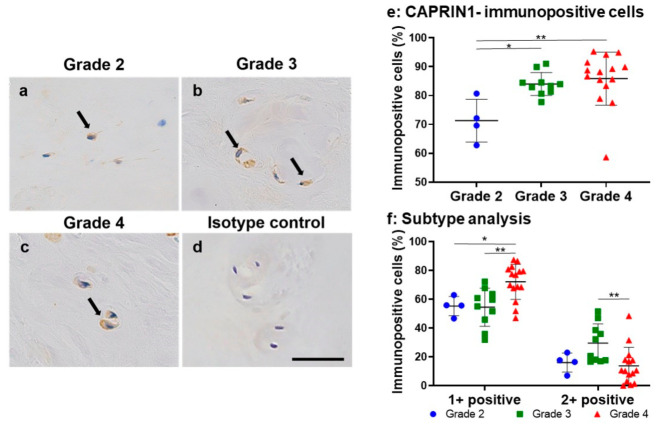
Immunohistochemical staining for CAPRIN1 (**a**–**c**) in human nucleus pulposus (NP) tissues. Pfirrmann grade 2 (**a**), grade 3 (**b**), and grade 4 (**c**), and isotype control (**d**). Scale bar: 50 μm (×400 magnification). Percentage of immunopositive cells for CAPRIN1 (**e**) and the subtype analysis by the intensity of immunostaining (**f**) in human nucleus pulposus (NP) tissues by Pfirrmann classification. * *p* < 0.05, ** *p* < 0.01.

**Figure 9 ijms-24-05768-f009:**
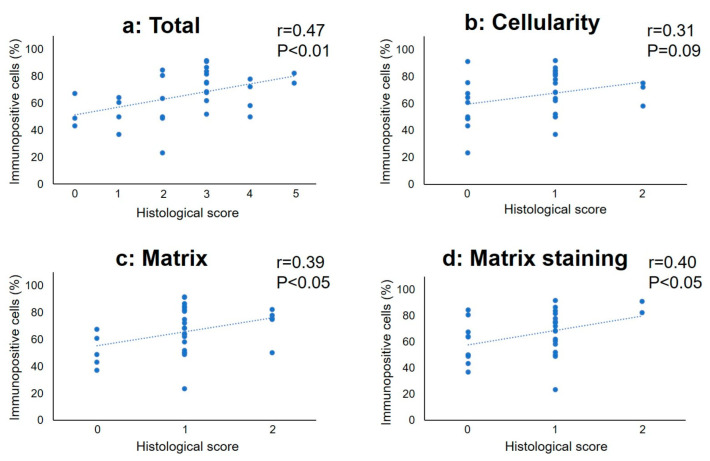
Correlation between the percentage of immunopositive cells for GADD45G and histological score in the total (**a**), cellularity (**b**), matrix (**c**), and (**d**) matrix staining score. r: correlation efficient.

**Figure 10 ijms-24-05768-f010:**
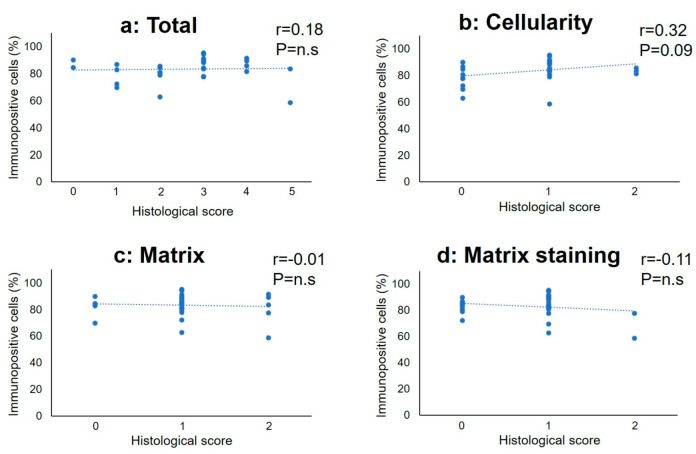
Correlation between the percentage of immunopositive cells for CAPRIN1 and histological score in the total (**a**), cellularity (**b**), matrix (**c**), and (**d**) matrix staining score. r: correlation efficient.

**Figure 11 ijms-24-05768-f011:**
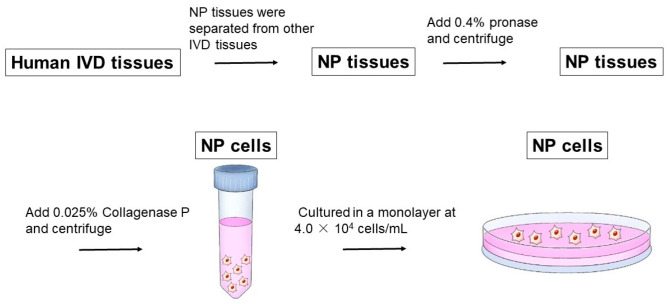
Isolation of the human nucleus pulposus (NP) cells.

**Table 1 ijms-24-05768-t001:** Information on patients and intervertebral discs used in this study.

Patient #	Age (Year-Old)	Gender	MRI Grade	Disease
1	6X	Male	3	LSS
2	4X	Male	3	LSS
3	6X	Male	3	DH
4	7X	Female	3	LSS
5	6X	Male	3	LSS
6	6X	Female	3	LSS
7	4X	Male	2	Spinal metastasis
8	3X	Male	2	Trauma
9	3X	Male	2	Trauma
10	3X	Male	2	Trauma
11	5X	Female	3	LSS
12	6X	Male	3	LSS
13	7X	Female	3	LSS
14	6X	Female	3	LSS
15	6X	Female	3	LSS
16	7X	Female	4	LSS
17	7X	Female	4	LSS
18	6X	Female	4	DH
19	7X	Female	4	LSS
20	8X	Female	4	LSS
21	7X	Male	4	DH
22	6X	Female	4	LSS
23	7X	Female	4	LSS
24	8X	Female	4	LSS
25	7X	Male	4	LSS
26	7X	Female	4	LSS
27	6X	Male	4	DH
28	7X	Male	4	LSS
29	6X	Male	4	LSS
30	7X	Female	4	LSS

LSS: lumbar spinal stenosis, DH: disc herniation.

**Table 2 ijms-24-05768-t002:** Classification of immunopositive cells.

		Staining Area		
		Under Half of the Nucleus	Half of the Nucleus	Over Half of the Nucleus
**Staining intensity**	Low	1+	1+	2+
	High	1+	2+	2+

## Data Availability

The data presented in this study are available upon reasonable request from the corresponding authors.

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
