# Peer review of "Expression of GADD45G and CAPRIN1 in Human Nucleus Pulposus: Implications for Intervertebral Disc Degeneration"

_ijms, 2023, doi:10.3390/ijms24065768_

Round 1
Reviewer 1 Report
The qPCR primer sequences should be provided.
For western blot and IHC, kindly specify the antibody supplier and dilution factor.
Why did the authors only examine the IL-1β why not other proinflammatory cytokines?
The author should revise the H&E and safranin O staining. Please mention the scale bar and provide the 10 and 20x magnification for better understanding.
Why the author does not utilize arrows to make it easier to distinguish the results of the H&E and safranin O stains.
It is quite difficult to observe the results of safranin O, so the author should provide a better magnification photograph.
The discovery, in my perspective, lacks sufficient information to say implications for intervertebral disc degeneration.
The authors carefully reviewed the figures because there are some spelling errors in the figure labeling.
Fig 5. Is there possible please add an experimental control group and please quantify the results. Fig.5 (a) the authors should provide better blots.
Don't include generic information in the introduction. The introduction and discussion should be revised by the authors, as I feel some parts are duplicated.
Author Response
Reviewer 1
Response) The authors sincerely appreciate the reviewer taking the time to review our manuscript and giving us thoughtful comments.
- The qPCR primer sequences should be provided.
Response) The qPCR primer sequences have not been disclosed by the supplier; therefore, the authors can’t provide them. The authors appreciate your understanding.
- For western blot and IHC, kindly specify the antibody supplier and dilution factor.
Response) According to the reviewer’s comment, the supplier of all the antibodies was described, and the following descriptions were added to the Materials and Methods section.
Line 350:
diluted with 5% skim milk in Tris Buffered Saline (TBS)
Line 402 to 403:
(Proteintech, Manchester, UK) diluted with 1% bovine serum albumin (BSA) in Phosphate Buffered Saline (PBS)
- Why did the authors only examine the IL-1β why not other proinflammatory cytokines?
Response) Thank you for the reviewer’s important comment. The authors previously examined the inflammatory effect of IL-1β in vitro, and the stable inflammatory effect was confirmed by IL-1β (the following article). Thus, IL-1β was used as the representative of proinflammatory cytokines.
Sano, T., K. Akeda, J. Yamada, N. Takegami, T. Sudo, and A. Sudo. "Expression of the Rank/Rankl/Opg System in the Human Intervertebral Disc: Implication for the Pathogenesis of Intervertebral Disc Degeneration." BMC Musculoskelet Disord 20, no. 1 (2019): 225.
- The author should revise the H&E and safranin O staining. Please mention the scale bar and provide the 10 and 20x magnification for better understanding.
Response) Thank you for the reviewer’s important suggestion. According to the reviewer’s comment, the authors revised the H&E and safranin O staining (Figures 1 and 2) and added 200x magnification figures (Figure 1a-c and Figure 2a-c).
- Why the author does not utilize arrows to make it easier to distinguish the results of the H&E and safranin O stains.
Response) Thank you for the reviewer’s important comment. The arrows, arrowheads, and asterisks were added to the H&E and safranin O staining (Figures 1 and 2).
- It is quite difficult to observe the results of safranin O, so the author should provide a better magnification photograph.
Response) Thank you for the reviewer’s important comment. According to the reviewer’s comment, the authors revise the safranin O staining (Figure 2) and added 200x magnification figures (Figure 2a-c).
- The discovery, in my perspective, lacks sufficient information to say implications for intervertebral disc degeneration.
Response) The authors agree with the reviewer’s notion. Therefore, the descriptions in the Abstract and Conclusion sections were revised as follows.
Lines 28 to 30:
suggesting that it may be regulated during the progression of IVD degeneration to maintain the integrity of human NP tissues by controlling cell proliferation and apoptosis under epigenetic alteration.
Lines 431 to 434:
The authors speculated that the expression of these two cell cycle–related molecules may be regulated during IVD degeneration progression to maintain the integrity of human NP tissues by controlling cell proliferation and apoptosis (cell death) under epigenetic alteration.
- The authors carefully reviewed the figures because there are some spelling errors in the figure labeling.
Response) It was corrected accordingly.
- Fig 5. Is there possible please add an experimental control group and please quantify the results. Fig.5 (a) the authors should provide better blots.
Response) Thank you for the reviewer’s important suggestion. Unfortunately, we have not examined other types of cells as an experimental control group. Fig.5 (a) was replaced with better blots, which were pretreated by chondroitinase ABC, and full blot images were shown. The following descriptions were added to the Materials and Methods section.
Lines 346 to 347:
For GADD45G-immunoblotting, the cell lysates were treated with chondroitinase-ABC, degrading the proteoglycans' chondroitin sulfate chains.
- Don't include generic information in the introduction. The introduction and discussion should be revised by the authors, as I feel some parts are duplicated.
Response) According to the reviewer’s comment, the following descriptions in the Introduction and Discussion sections were deleted.
Lines 41 to 43 (original)
The human vertebral column complex consists of ventrally located vertebral bodies and intervening IVDs that provide movement between adjacent vertebral bodies, absorb shock, and transmit loads through the vertebral column [8].
Lines 63 to 69 (original)
Cell cycle is divided into a synthesis phase (S) and a mitotic segregation phase (M), with two intervenient gap phases (G1 and G2) preceding the S and M phases. Cells already entering the cell cycle are further controlled by three checkpoints: the G1/S, G2/M, and mitotic spindle checkpoints [13]. This cell cycle or checkpoints regulated by molecules such as cyclin, cyclin-dependent kinases (CDKs), and CDK inhibitors (CKIs) [14]. Recently, Cui et al. [15] reported that S-phase kinase-associated protein-2 (Skp2), one of cell cycle-associated proteins, promoted cell cycle by CDK2 activation in NP cells.
Lines 253 to 255 (original)
The GADD45 family has been shown to play essential roles in cell cycle arrest, DNA repair, cell survival, and apoptosis in response to environmental and physiological stress (see review in [18, 19]).
Lines 302 to 304 (original)
In a previous study, we evaluated the association between DNA methylation and disc degeneration using the genome-wide association analysis of human IVD tissues and identified 220 differentially methylated loci associated with IVD degeneration.
Reviewer 2 Report
This is an interesting study investigating the expression of 2 genes, GADD45G and CAPRIN1, which were identified from a previous genome wide methylation study in human intervertebral disc degeneration.
Major comments:
1. Fig. 5, please provide full blot images.
2. Please provide clinical information in tabular format, e.g. the breakdown of demographic and clinical profiles for different Pfirrmann grades.
Minor comments:
1. Could the authors clarify why for NP culture (Method 4.1), only Pfirrmann grade 4, and elderly female patients, were selected?
2. Please elaborate objective measure that led to the distinction between 1+ and 2+ for immunohistochemistry.
3. Could the authors clarify the selection of IL-1b concentration tested?
Author Response
Reviewer 2
Comments and Suggestions for Authors
This is an interesting study investigating the expression of 2 genes, GADD45G and CAPRIN1, which were identified from a previous genome wide methylation study in human intervertebral disc degeneration.
Response) Thank you very much for the reviewer’s kind comments and understanding of our manuscript.
Major comments:
- Fig. 5, please provide full blot images.
Response) It was corrected accordingly.
- Please provide clinical information in tabular format, e.g. the breakdown of demographic and clinical profiles for different Pfirrmann grades.
Response) Thank you for the reviewer’s important suggestion. According to the reviewer’s comment, the following table was added to the Introduction section as Table 1.
Table 1. Information on patients and intervertebral discs used in this study
|
Patient # |
Age (years-old) |
Gender |
MRI Grade |
Disease |
|
1 |
6X |
Male |
3 |
LSS |
|
2 |
4X |
Male |
3 |
LSS |
|
3 |
6X |
Male |
3 |
DH |
|
4 |
7X |
Female |
3 |
LSS |
|
5 |
6X |
Male |
3 |
LSS |
|
6 |
6X |
Female |
3 |
LSS |
|
7 |
4X |
Male |
2 |
Spinal metastasis |
|
8 |
3X |
Male |
2 |
Trauma |
|
9 |
3X |
Male |
2 |
Trauma |
|
10 |
3X |
Male |
2 |
Trauma |
|
11 |
5X |
Female |
3 |
LSS |
|
12 |
6X |
Male |
3 |
LSS |
|
13 |
7X |
Female |
3 |
LSS |
|
14 |
6X |
Female |
3 |
LSS |
|
15 |
6X |
Female |
3 |
LSS |
|
16 |
7X |
Female |
4 |
LSS |
|
17 |
7X |
Female |
4 |
LSS |
|
18 |
6X |
Female |
4 |
DH |
|
19 |
7X |
Female |
4 |
LSS |
|
20 |
8X |
Female |
4 |
LSS |
|
21 |
7X |
Male |
4 |
DH |
|
22 |
6X |
Female |
4 |
LSS |
|
23 |
7X |
Female |
4 |
ASD |
|
24 |
8X |
Female |
4 |
LSS |
|
25 |
7X |
Male |
4 |
LSS |
|
26 |
7X |
Female |
4 |
LSS |
|
27 |
6X |
Male |
4 |
DH |
|
28 |
7X |
Male |
4 |
LSS |
|
29 |
6X |
Male |
4 |
LSS |
|
30 |
7X |
Female |
4 |
LSS |
LSS: Lumbar Spinal Stenosis, DH: Disc Herniation.
Minor comments:
- Could the authors clarify why for NP culture (Method 4.1), only Pfirrmann grade 4, and elderly female patients, were selected?
Response) Thank you for the reviewer’s critical comment. Human NP tissues for in vitro studies were obtained from consecutive lumbar surgeries. Therefore, the authors did not intentionally select gender or degenerative degree.
- Please elaborate objective measure that led to the distinction between 1+ and 2+ for immunohistochemistry.
Response) Thank you for pointing out an important issue. The following scoring system was used to determine +1 or +2 staining. Therefore, the following table was added to the Materials and Methods section.
Table 2. Classification of immunopositive cells.
|
|
Staining area |
|||
|
Under half of the nucleus |
Half of the nucleus |
Over half of the nucleus |
||
|
Staining intensity |
Low |
1+ |
1+ |
2+ |
|
High |
1+ |
2+ |
2+ |
|
- Could the authors clarify the selection of IL-1b concentration tested?
Response) Thank you for the reviewer’s important comment. The authors previously examined the inflammatory effect of IL-1β on human IVD cells in vitro by this concentration, and stable and dose-dependent inflammatory effects have been confirmed (following article). Thus, this concentration was selected in this study.
Sano, T., K. Akeda, J. Yamada, N. Takegami, T. Sudo, and A. Sudo. "Expression of the Rank/Rankl/Opg System in the Human Intervertebral Disc: Implication for the Pathogenesis of Intervertebral Disc Degeneration." BMC Musculoskelet Disord 20, no. 1 (2019): 225.